# Using Residual Dynamic Structural Equation Modeling to Explore the Relationships among Employees’ Self-Reported Health, Daily Positive Mood, and Daily Emotional Exhaustion

**DOI:** 10.3390/healthcare9010093

**Published:** 2021-01-18

**Authors:** Ya-Tzu Kung, Shyh-Ching Chi, Yung-Chou Chen, Chia-Ming Chang

**Affiliations:** 1Graduate Institute of Physical Education, National Taiwan Sport University, Taoyuan 33301, Taiwan; ivy022011@gmail.com; 2Department of Sports Training Science-Balls, National Taiwan Sport University, Taoyuan 33301, Taiwan; cn6166@ntsu.edu.tw; 3Department of Athletic Sports, National Chung Cheng University, Chiayi 62103, Taiwan; enzochen@ccu.edu.tw; 4Department of Physical Education, Health & Recreation, National Chiayi University, Chiayi 62103, Taiwan

**Keywords:** residual dynamic, structural equation modeling, daily positive mood, daily emotional exhaustion, self-reported health

## Abstract

This study examined the relationships among self-reported health, daily positive mood, and daily emotional exhaustion among employees in health and fitness clubs using residual dynamic structural equation modeling (RDSEM). A questionnaire was completed by 179 employees at recruitment and then a diary survey over 10 consecutive workdays. Results of RDSEM analyses revealed that daily positive mood was negatively associated with daily emotional exhaustion at both within-person and between-person levels. Self-reported health was positively related to the person’s mean of daily positive mood and negatively associated with the person’s mean of daily emotional exhaustion. Self-reported health moderated the relationship between daily positive mood and daily emotional exhaustion; employees with higher self-reported health levels tend to respond with larger changes in their daily emotional exhaustion when their daily positive mood changes. These findings provide important insights for organizations aiming at their employees’ health, happiness, and job burnout.

## 1. Introduction

Emotional exhaustion is defined as the feeling of physical and emotional depletion and is recognized as the most important component of job burnout [1,2]. Job burnout is a state of mental and physical exhaustion caused by one’s professional life [3] and is probably one of the most popular research topics in occupational health psychology. Researchers have persuasively demonstrated that employees who are at risk of burnout show impaired job performance and various job withdrawal behaviors—absenteeism, intention to leave the job, and actual turnover [4,5].

Basinska and Gruszczynska [6] claimed that the traditional research approach failed to recognize the chronic, enduring, and dynamic nature of job burnout, therefore longitudinal studies were preferred. However, there is a new approach regarding job burnout as also being changeable over a shorter time frame [7]. This approach could be called as state approach assuming that emotional exhaustion or burnout experiences may vary within the same employee from one moment or day to another as a response to specific events at work [6,7]. Therefore, job burnout as indicated by emotional exhaustion can be conducted in a daily diary method.

Moods can provide the affective context for thought processes and capture typical day-to-day feeling states in contrast to more extreme emotional experiences [8]. According to the enrichment model proposed by Greenhaus and Powell [9], a person’s mood may act as a resource that is transferred to another life domain and lead to life domain enrichment. Edwards and Rothbard [10] asserted that a positive mood can improve task activity, enhance cognitive functioning, and promote positive interactions with others. In addition, the broaden-and-build theory proposes that positive affect can broaden thought-action repertoire and facilitate the use of personal resources [11]. Thus, positive mood plays a potentially important role in the context of the workplace. Studies have found that positive moods encourage higher quality service [12] and produce more innovative and flexible solutions to problems [13].

Lyubomirsky, King, and Diener [14] argued that the propensity to frequently experience positive mood has been related to successes in everyday life. Clark and Watson [15] used a diary method to assess mood and social activity in a sample of young adults and found that a high positive affect was associated with reported social activity, especially physical activity. Therefore, a positive mood could be daily personal resources influencing individual feelings and behavior at work [16]. Based on these arguments, it is hypothesized that daily positive mood at the beginning of work is negatively related to daily emotional exhaustion after work.

There is a growing literature showing that emotional exhaustion and burnout were linked to poor health through various negative health behaviors, such as sleep disturbance, and through other physiological and biological mechanisms [5,17]. Vinokur, Pierce, and Lewandowski-Romps [18] argued that emotional exhaustion and general health were related to each other. That is, emotional exhaustion impairs individual health and the depletion of individual health leads to emotional exhaustion. According to the conservation of resources (COR) theory, people use important resources to respond to stress and construct a reservoir of sustaining resources for times of future need [19]. Hobfoll, Halbesleben, Neveu, and Westman [19] mentioned that commonly valued resources are health, well-being, family, self-esteem, and a sense of purpose and meaning in life. There is reason to believe that emotional exhaustion represents a process involving the depletion of resources in the workplace [19,20].

In light of the later formulations of the job demands–resources (JD–R) model, personal resources have been put into the model and regarded as that they may have similar motivational potential to that of job resources and may be positively related to work engagement, and consequently to positive work-related outcomes [21]. Personal resources are, therefore, an important factor in facilitating adaptation to work environment [22], coping with highly demanding work conditions [23], promoting worker well-being [24], and preventing exhaustion and burnout [25]. Previous studies with personal resources integrated with the JD–R model focused on cognitive aspects, such as self-efficacy, self-esteem, or optimism [26]. This may be a limitation because physical and social aspects were also found to be important [27]. Health is a resource for daily living [28] and a positive concept emphasizing personal resources and physical capacities for everyday life. Shirom, Melamed, Toker, Berliner, and Shapira [29] contended that good health as indexed by self-reported health (SRH) should be negatively associated with burnout because it represents a pivotal coping resource and any changes in it are prone to influence one’s level of burnout. The longitudinal study of Vinokur, Pierce, and Lewandowski-Romps [18] was able to illustrate this perspective. Their findings showed perceived health, as measured in 2001, was negatively related to job burnout as measured in 2004.

During the past decades, numerous researchers found evidence regarding the physical health benefits of positive affect [14,30,31,32,33,34]. As Lyubomirsky et al. [14] have stated, not surprisingly, high positive affect and low negative affect have also been associated with subjective reports of better health. However, Finch, Baranik, Liu, and West [35] found that self-reports of physical health were linked to subsequent positive affect, but that positive affect was not related to subsequent reports of physical health. This finding challenges those research studies focusing on positive affect as a predictor of physical health.

There is a lot of literature studying the impact of emotional exhaustion or positive affect on health. Only very few articles focused on the contributions of health to emotional exhaustion [18,36,37] and affect [35,38]. Thus, our study aims to fill these research gaps and hypothesized that employees’ self-reported health is positively related to their daily positive mood and negatively related to their daily emotional exhaustion.

Based on the studies from the JD–R model, Xanthopoulou, Bakker, Demerouti, and Schaufeli [21] argued that personal resources could moderate the relationship between job demands and emotional exhaustion. Schmitz, McCluney, Sonnega, and Hicken [39] asserted that subsequent versions of the JD–R model have been expanded to include personal resources (e.g., perceived mental and physical health), which are supposed to increase involvement and mitigate the relationship between job demands and burnout. Brenninkmeijer, Demerouti, le Blanc, and van Emmerik [40] found that personal resources significantly moderated the effects of job characteristics on teachers’ emotional exhaustion and work engagement. In addition, Schaufeli and Taris [41] claimed that the JD–R model is heuristic in nature and provided different pathways that personal resources may be integrated into the JD–R model. Zellars, Perrewé, Hochwarter, and Anderson [42] used the COR framework to explore the interactive effects of two personal resources on strain. Their findings indicated that conscientiousness moderates the relation between positive affect and burnout. Based on the moderating assumptions of JD–R model, COR framework, and pathways identified by Schaufeli and Taris [41], our study intends to consider the interactive effects of different personal resources (mental and physical aspects) on the ill-being of employees and propose that personal resource of physical aspect may moderate the relationship between personal resources of mental aspect and exhaustion. Therefore, it is hypothesized that self-reported health moderates the relation between daily positive mood and daily emotional exhaustion.

The purpose of the present study was to explore the relationships among employees’ self-reported health, daily positive mood, and daily emotional exhaustion in health and fitness clubs. To examine the hypotheses derived from a daily diary data structure, we employed a novel statistical technique created by Asparouhov and Muthén [43]—residual dynamic structural equation modeling (RDSEM). It allows us to study both observed and latent variables using dairy data with observations from multiple individuals collected at many time points [43]. Therefore, RDSEM permits estimating within-person auto-regression between the residuals for daily emotional exhaustion and within-person regression effect of daily emotional exhaustion on daily positive mood by including time as a time-varying covariate in the model. In the between-level, self-reported health, as a time-invariant covariate, could be added to the model to predict intercepts of daily emotional exhaustion and daily positive mood and regression slopes of daily emotional exhaustion on daily positive mood. In addition, it also permits examining the between-person effect between daily positive mood and daily emotional exhaustion.

## 2. Method

### 2.1. Study Design and Participants

Participants in this study were employees working for the health and fitness clubs located in different cities in Taiwan. A total of 200 employees were recruited to participate in our study. Data were collected using the 10-day diary method and under a clearly informed consent agreement. The research was reviewed and approved by the Institutional Review Board. For the study, 179 employees completed surveys using paper questionnaires over 10 days. The within-person measures (i.e., daily positive mood, daily emotional exhaustion) response rate was 89.5%. Among 179 participants, 68.15% (*n* = 122) were identified as women, and 77.09% (*n* = 138) were married. The average age was 30.42. The average job tenure was 5.29 years.

### 2.2. Research Instruments

Research participants were invited to complete their 10-day diary. Positive mood and emotional exhaustion were evaluated on a daily basis; thus, the data were retrieved for 10 consecutive days. On the other hand, self-reported health was assessed on the first day of participation. The measures used in the research are presented as follows.

### 2.3. Day-Level Emotional Exhaustion Scale

The four statements of the day-level emotional exhaustion scale were modified from the measurement of experienced emotional exhaustion originally introduced by Maslach and Jackson [44]. Statements were rewritten to best suit our study purpose. The four statements were (a) I feel burned out from my work; (b) dealing with customers all day is really a strain for me; (c) I feel used up at the end of the workday; and (d) I feel emotionally drained from work today [45]. All items were given on a five-point Likert scale (1 = strongly disagree, 5 = strongly agree). Chang, Liu, Huang, and Hsieh [45] had examined the validity and reliability of this scale. The fit measures of confirmatory factor analysis indicated the well-fitting model (χ^2^/df = 3.33, RMSEA (root mean square error of approximation) = 0.07, GFI (goodness of fit index) = 0.98, CFI (comparative fit index) = 0.98, NNFI (non-normed fit index) = 0.95). The reliability of the scale was good (Cronbach’s α = 0.90) [45].

### 2.4. Self-Reported Health (SRH)

The perception of employees about their health status was collected through the single-item question, “You would say your health is …?” Answers were given through selection and on a five-point scale (very bad, bad, fair, good, and very good).

### 2.5. Day-Level Positive Mood

Day-level positive mood used an item developed from Xanthopoulou, Bakker, Demerouti, and Schaufeli [46]. It recalled from the beginning of the workday (“Today, I came to work in a very pleasant mood”). The item was measured on a five-point scale (1 = very disagree, 5 = very agree).

### 2.6. Data Analytic Strategy

The study analyzed the dynamic picture constructed from the data of employees’ self-rated health, 10-day day-level emotional exhaustion, and 10-day day-level positive mood. In statistical analytic strategy, residual dynamic structural equation models (RDSEM) was used. RDSEM is a variant of DSEM, which combines four different modeling techniques—multilevel modeling, time-series modeling, structural equation modeling (SEM), and time-varying effects modeling (TVEM). The RDSEM framework is mainly based on the traditions of the SEM approach and could be applied to study factor analysis, path analysis, mediation and moderation analysis, and their evolution across time [43]. Moreover, RDSEM is better than DSEM to model time trends. While time trend is taking into consideration in DSEM, the estimates recover the coefficients accurately but not the variance terms [47].

The path diagram representation of our RDSEM is presented in Figure 1. We followed the conventions from Curran and Bauer [48] for visually representing models with multiple levels, the intercept, the lag-1 slope, and the time-varying covariate slope that all have superimposed circles.

In equation form, our model is
$$Exhause_{ti}=\alpha_{i}+\beta_{1i}Mood_{ti}^{c}+\beta_{2i}Time_{ti}+e_{ti}$$
$$e_{ti}=\varphi_{i}e_{\left(t-1\right)i}+\delta_{ti}$$
$$\alpha_{i}=\gamma_{00}+\gamma_{01}Health_{i}^{g}+\gamma_{02}Mood_{i}^{b}+\mu_{0i}$$
(1)$$\varphi_{i}=\gamma_{10}+\mu_{1i}$$
$$\beta_{1i}=\gamma_{20}+\gamma_{21}Health_{i}^{g}+\mu_{2i}$$
$$\beta_{2i}=\gamma_{30}+\mu_{3i}$$
$$Mood_{i}^{b}=\gamma_{40}+\gamma_{41}Health_{i}^{g}+\mu_{4i}$$
where the *c* superscript on $Mood_{ti}^{c}\;$ indicates that the covariate is person-mean centered; the *g* superscript on $Health_{i}^{g}$ indicates that it has been grand-mean centered; $Mood_{i}^{b}$ is between-level components that is latent variable with an estimated mean and variance; $\sigma_{i}^{2}$ is the residual variance of emotional exhaustion that allows the difference for each person in the data; $\gamma_{00},\;\;\gamma_{10},\;\gamma_{20},\;\gamma_{30},\;\gamma_{50}$ and $\mu_{0i},\;\mu_{1i},\;\mu_{2i},\;\mu_{3i},\;\;\mu_{5i}\;$ are the intercepts and variances for $\alpha_{i},\;\varphi_{i},\;\beta_{1i},\;\beta_{2i}\;$ and $Mood_{i}^{b}$, respectively; and $\gamma_{01},\;\gamma_{21}\;$ and $\gamma_{51}$ are the regression coefficients of $\alpha_{i},\;\beta_{1i}$ and $Mood_{i}^{b}\;$ on $Health_{i}^{g}.$

## 3. Results

Descriptive results appear in Table 1. Included in the table, the pooled within-person correlations and the sample size weighted between-person correlations were calculated using stats function of psych package in R language. Each of the reported correlations is in the expected direction.

The result of estimates and 95% credible intervals for RDSEM of daily emotional exhaustion is presented in Table 2. While controlling for the linear trend (Intercept(beta2i) = −0.020, non-null), the average horizontal mean line across all employees around which daily emotional exhaustion varies is 4.743, and there is non-null variability around this line (0.093), meaning that some employees have consistently higher or lower horizontal mean lines than the average. The average residual autoregressive coefficient across all employees is 0.281, and non-null showing that residuals of daily emotional exhaustion have stability. Regarding the effect of daily positive mood, the average effect across all employees is –0.444 and no-null, meaning that a one-unit increase in daily positive mood predicts a decrease in the employee’s emotional exhaustion horizontal mean line of 0.444 points. That is, daily positive mood is negatively related to daily emotional exhaustion.

The effects of self-reported health on a person’s means of daily emotional exhaustion and daily positive mood were non-null ($\gamma_{01}$= −0.173; $\gamma_{41}$= 0.148), meaning that a one-unit increase in self-reported health predicts a decrease in the daily emotional exhaustion horizontal mean line of 0.173 points and an increase in daily positive mood horizontal mean line of 0.148 points. The effect of self-reported health on beta1i was positive and non-null (0.186), meaning that employees with a higher self-reported health level tend to respond with larger changes in their daily emotional exhaustion when their daily positive mood changes. These results illustrated that self-reported health has a moderating effect on the relationship between daily positive mood and daily emotional exhaustion.

## 4. Discussion

### 4.1. Effects of Daily Positive Mood on Daily Emotional Exhaustion

The daily diary method was used to collect 10-day repeated measures of daily positive mood and daily emotional exhaustion from employees at health and fitness clubs. Using residual dynamic structural equation modeling, we can examine within- and between-person variations with regard to both daily positive mood and daily emotional exhaustion. To date, researchers studying the relationships between affect and burnout in the workplace have primarily focused on between-person associations, sometimes neglecting within-person processes [7,49].

With regard to the within-person variance, the result of this study showed that daily positive mood was negatively associated with daily emotional exhaustion. That is, the more positive mood employees feel at the beginning of the day, the lower their degrees of daily emotional exhaustion throughout that day. In the between-person variance, the result also showed that a person’s mean of daily positive mood negatively predicts the person’s mean of daily emotional exhaustion.

According to the enrichment model, broaden-and-build theory, and COR theory, resources play a potentially important role in the context of the workplace. The more resources individuals have available to them, the more likely they are prevented from experienced stress [50]. Lyubomirsky, King, and Diener [14] argued that experiencing a positive mood has been related to successes in everyday life. Employees’ moods at the start of the workday influenced how they felt and performed for the rest of the day [51]. Positive affect has been typically associated with concurrent decreases in strain and increases in health and well-being [52]. Our study provides support for these theoretical and practical arguments in the workplace and on a day-to-day basis.

### 4.2. Effects of Self-Reported Health on Positive Mood and Emotional Exhaustion

The second goal of our study was to investigate the effects of self-reported health on the daily positive mood and daily emotional exhaustion. Our residual dynamic structural equation model allowed the partitioning of the variances of daily positive mood and daily emotional exhaustion into the between-person variances so that we can set the relationships among self-reported health, daily positive mood, and daily emotional exhaustion in the between level. The interpretation of the structural coefficients in the between level is the same as in the standard structural equation model. The results of our study revealed that self-reported health was positively related to the person’s mean of daily positive mood and negatively associated with the person’s mean of daily emotional exhaustion.

These findings support our understanding of the underlying mechanism that guides the personal resource process of the COR theory. Health is a resource for daily living [28] and a positive concept emphasizing personal resources and physical capacities for everyday life. Shirom, Melamed, Toker, Berliner, and Shapira [29] contended that good health as indexed by SRH should be negatively associated with burnout because it represents a pivotal coping resource, and any changes in it are prone to have an impact on one’s level of burnout. Health serves as a global resource needed for normal functioning in nearly all life domains [18]. Therefore, good health is linked to positive affect [35], while poor health is indicative of chronic stress [53] or depletion of specific resources needed to combat job burnout.

### 4.3. The Moderating Role of Self-Reported Health

The third goal of our study aimed to uncover the moderating role of self-reported health, as a personal resource, on the relationship between daily positive mood and daily emotional exhaustion. Based on the moderating assumptions of the JD–R approach, COR framework, and pathways identified by Schaufeli and Taris [41], we provided an interactive pathway examining two personal resources on daily emotional exhaustion. Our findings presented that self-reported health has a moderating effect on the relationship between daily positive mood and daily emotional exhaustion. Employees with higher self-reported health status tend to respond with larger changes in their daily emotional exhaustion when their daily positive mood changes. Thus, the interactive effects of the two personal resources on emotional exhaustion hold.

Cropanzano and Wright [54] asserted that happy workers are a valuable asset for organizations; our study confirmed another valuable asset for organizations—the physical health of employees. In addition, health, including psychological and physical aspects, is an important personal resource for employees. Our study also confirmed that positive mood at the beginning of the workday has a reverse relationship with daily emotional exhaustion. This demonstrated the importance of variation in state affect, something that organizations are able to influence each workday. Therefore, we encourage managers of health and fitness clubs to be aware of their role in setting the start-of-workday mood. Apart from this, health and fitness clubs know how to improve health for their customers. They should use these ways to enhance the health of their employees.

Despite obtaining interesting results, some limitations need to be acknowledged when interpreting the results of this study. Due to the study variables solely obtained from self-reports, common method variances need to be concerned. Common method variance might cause research finding bias by inflating the examined relationships [55]. The results may also not be generalizable to the broader population of employees of health and fitness clubs due to the use of a convenience sample. Participants were prone to be employees of health and fitness clubs who identified with feeling burned out at work or had a specific interest in burnout.

## 5. Conclusions

### 5.1. Summary

The purpose of this study was to examine the relationships among employees’ self-reported health, daily positive, and daily emotional exhaustion in health and fitness clubs. In this study, we adopted a new technique to combine both stable and trait approaches to look into whether chronic enduring may lead to emotional exhaustion. The daily diary method was used to collect longitudinal data, and residual dynamic structural equation modeling was used to test the hypothesized research theories.

The study showed employees’ daily positive mood and daily emotional exhaustion were negatively correlated, and employees’ self-reported health was positively related to the average positive mood. In addition, employees’ self-reported health was negatively related to the average emotional exhaustion. As for the moderation effects, self-reported health moderated employees’ average positive mood and emotional exhaustion. Employees with higher self-reported health status tend to respond with larger changes in their daily emotional exhaustion when their daily positive mood changes.

### 5.2. Implication

The results of this study suggest the human resources department think highly of their employees’ health status and create a positive work atmosphere to reduce employees’ burnout and promote work efficacy.

## Figures and Tables

**Figure 1 healthcare-09-00093-f001:**
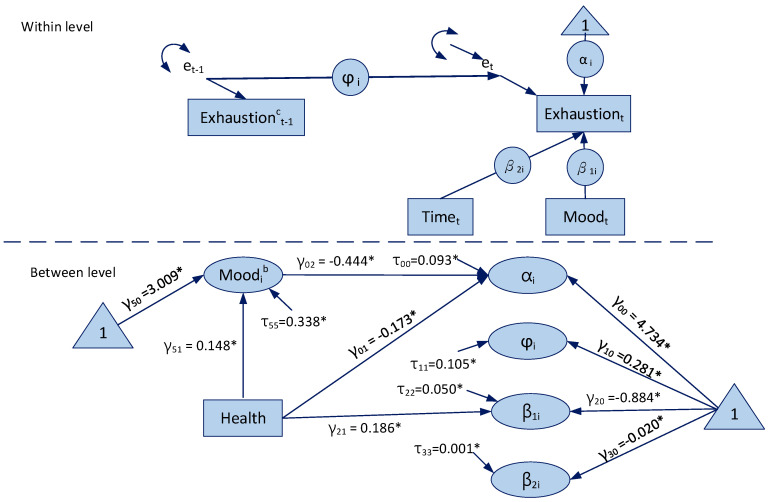
Path diagram and estimated parameters of the study model. * *p* < 0.05.

**Table 1 healthcare-09-00093-t001:** Means, standard deviations, and correlations between the study variables.

	Mean	SD	1	2	3
1. Self-reported health	3.464	0.689	1.000	NA	NA
2. Day-level positive mood	3.530	0.655	0.160 *	1.000	−0.273 ***
3. Day-level emotional exhaustion	2.490	0.595	−0.303 ***	−0.393 ***	1.000

Note: *N* = 179 employees over *N* = 1790 occasions. Day-level data was averaged across 10 days. Relationships below the diagonal are the sample size weighted between-person correlations; relationships above the diagonal are the pooled within-person correlations. *** *p* < 0.001; * *p* < 0.05; NA is not available.

**Table 2 healthcare-09-00093-t002:** Estimates and 95% credible intervals for residual dynamic structural equation modeling (RDSEM) of emotional exhaustion with a linear trend.

Effect	Notion	Posterior Median	95% Credible Interval
Intercept(alpha)	$\gamma_{00}$	4.743 *	(4.259, 5.257)
Intercept (phi)	$\gamma_{10}$	0.281 *	(0.199, 0.377)
Intercept (beta1i)	$\gamma_{20}$	−0.884 *	(−1.134, −0.623)
Intercept (beta2i)	$\gamma_{30}$	−0.020 *	(−0.029, −0.011)
Intercept (Mood)	$\gamma_{40}$	3.009 *	(2.500, 3.514)
Alpha on health	$\gamma_{01}$	−0.173 *	(−0.272, −0.073)
Alpha on Moodib	$\gamma_{02}$	−0.444 *	(−0.587, −0.305)
beta1i on Health	$\gamma_{21}$	0.186 *	(0.170, 0.255)
Moodib on Health	$\gamma_{41}$	0.148 *	(0.009, 0.295)
var.(phi)	$\tau_{11}$	0.105 *	(0.072, 0.150)
var.(beta2i)	$\tau_{33}$	0.001 *	(0.000, 0.001)
Res. Var.(alpha)	$\tau_{00}$	0.093 *	(0.050, 0.139)
Res. Var.(Moodib)	$\tau_{44}$	0.338 *	(0.258, 0.441)
Res. Var.(beta1i)	$\tau_{22}$	0.050 *	(0.030, 0.079)

* *p* < 0.05.

## Data Availability

Data sharing not applicable.

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
