# Peer review of "Using Residual Dynamic Structural Equation Modeling to Explore the Relationships among Employees’ Self-Reported Health, Daily Positive Mood, and Daily Emotional Exhaustion"

_healthcare, 2021, doi:10.3390/healthcare9010093_

Round 1

Reviewer 1 Report

• This is an interesting topic to improve the understanding regarding the health and happiness mood swings at job and considering job burnout. The authors do a good job to fetch a verity of data and integrate them to provide useful insights regarding the topic. I am in favor of this article as it contains several interesting components that should be published. However, I have following comments before final approval and publications. • Why RDSEM model was used and why not others? What are the main causes to prefer this over other? This can be answered in the response letter, no need to add into the paper at the moment. • Page 3, line 123-120, move this part to the methodology section. • How biasness was dealt in data collection? • What technique was used to collect data? Sampling method? • How to validate the data? • It would be better to begin the results section with some overview and general information of results and study rather starting with Table 1…………. • Results show some 95% of CI, however, methodology part is silent about that. What and how CI was adjusted? What was margin of error? And other statistical indicators? Please mention them in methodology. Afterwards, it will match to the results. • Moreover, results are too short with only 2 Tables presented. Explain them and add at least 2 figures. • Conclusion section should be different from the abstract, and it should be divided into two paragraphs. One containing the most significant results of the study and other with practical policy implications. • Keywords should be shortened as some keywords are way long such as “specially-designed green places/spaces within hotels.” And those should be no more than 6 usually. • First, only one figure is used and its quality should be improved. Second, is it possible to add more figures to balance the use of tables and figures? • I would suggest not to use citations older than 2015. In some cases, if it is damn necessary then try to use some classical ones. There are multiple references used from 1974, 1982, and 1991, etc. Please ponder them and consider replacement with the latest ones as much as possible.

Author Response

Reviewer 1

Comments and Suggestions for Authors

  • This is an interesting topic to improve the understanding regarding the health and happiness mood swings at job and considering job burnout. The authors do a good job to fetch a verity of data and integrate them to provide useful insights regarding the topic. I am in favor of this article as it contains several interesting components that should be published. However, I have following comments before final approval and publications.

Response: Thank you for comments.

Point 1:

Why RDSEM model was used and why not others? What are the main causes to prefer this over other? This can be answered in the response letter, no need to add into the paper at the moment.

Response 1:

Thank you for the sincere comments. The statistical approach RDSEM used in this research is the newest technique which can be applied to analyze the diary data, especially for longitudinal data. RDSEM is a variant of DSEM which combines four different modeling techniques: multilevel modeling, time-series modeling, structural equation modeling (SEM), and time-varying effects modeling (TVEM).

This study was designed to find the causal-effect relationships among proposed latent variables. Especially, the nature of the data contains multi-level dimensions and is time-dependent, so RDSEM was applied in the analysis. According to Asparouhov (2018), with data varied with time, RDSEM is better than DSEM to model time trend. For details in RDSEM, please refer to Asparouhov, T. (2018). Dynamic structural equation modeling of intensive longitudinal data using Mplus Version 8. Retrieved from http://www.statmodel.com/download/Part%205%20Asparouhov.pdf

Point 2:

Page 3, line 123-120, move this part to the methodology section.

Response 2:

Thank you for the cordial comments. We had move this in the method section. Please refer to lines 168-170.

Point 3:

How biasness was dealt in data collection?

Response 3:

Thank you for the earnest comments. Eighty percent of sports clubs are located in six major metropolitan cities in Taiwan. Therefore, we selected 13 clubs’ employees to participate in this study and we believe the sample can represent the population mostly and reduced biasness.

Point 4:

What technique was used to collect data?

Response 4:

Thank you for the comment. We use a structured questionnaire to collect data. The participants were invited to complete a 10- working days’ self-report evaluation sheets each day in rewards. Please refer to line 138.

Point 5:

Sampling method? •

Response 5:

Sampling method of this study was convenience sampling. In Taiwan, most sports clubs’ managers are familiar with YMCA’s secretaries. Therefore, we were able to ask YMCA’s secretaries to assist with the data collection. We then were able to recruit 200 participants from 13 sports clubs in Taiwan.

Point 6:

How to validate the data?

Response 6:

During the data collection period, we would send text messages to the participants every day as a reminder and out of the 200 participants, 179 were able to complete the 10-dairy questionnaires. After that, we assigned coding number to each questionnaire and carefully keyin the information. After that, we used descriptive statistics analyses to check if there were missing values and typing errors. We believe we had carefully validated the data collected.

Point 7:

It would be better to begin the results section with some overview and general information of results and study rather starting with Table 1…………. •

Response 7:

Thank you for the comment. We begin the section by “Descriptive results appear in Table 1. Included in the table….”, please refer to line 194. Thank you.

Point 8:

Results show some 95% of CI, however, methodology part is silent about that. What and how CI was adjusted? What was margin of error? And other statistical indicators? Please mention them in methodology. Afterwards, it will match to the results.

Response 8:

Thank you for the comment. RDSEM used Bayes Estimation which use 95% CI for significance test solely.Please refere to McNeish, D., & Hamaker, E. L. (2019, December 19). A Primer on Two-Level Dynamic Structural Equation Models for Intensive Longitudinal Data in Mplus. Psychological Methods. Advance online publication. http://dx.doi.org/10.1037/met0000250

Point 9:

Moreover, results are too short with only 2 Tables presented. Explain them and add at least 2 figures.

Response 9:

Thank you for the comment. This study used RDSEM to test the relationships among our proposed latent variables. We had looked up some similar studies used the same approach and adopted the same presentation style. We believe the table contains enough information needed for readers. Please refer to the following papers for detail. Thank you. 

Simbula, S. (2010). Daily fluctuations in teachers' well-being: A diary study using the Job Demands–Resources model. Anxiety, Stress, & Coping: An International Journal, 23(5), 563-584, DOI: 10.1080/10615801003728273

Breevaart, K., & Bakker, A. B. (2018). Daily job demands and employee work engagement: The role of daily transformational leadership behavior. Journal of Occupational Health Psychology, 23(3), 338–349. https://doi.org/10.1037/ocp0000082

Bakker, A. B., & Oerlemans, W. G. M. (2019). Daily job crafting and momentary work engagement: A self-determination and self-regulation perspective. Journal of Vocational Behavior, 112, 417-430.

Point 10:

Conclusion section should be different from the abstract, and it should be divided into two paragraphs.

One containing the most significant results of the study and other with practical policy implications.

Respons10: Thank you for the comment. We had revised the conclusion section. Please refer to lines 293-310.

Point 11:

Keywords should be shortened as some keywords are way long such as “specially-designed green places/spaces within hotels.” And those should be no more than 6 usually.

Response 11: The keywords used in this article are “residual dynamic; structural equation modeling; daily positive mood; daily emotional exhaustion; self-reported health” instead. Thank you.

Point 12:

First, only one figure is used and its quality should be improved.

Second, is it possible to add more figures to balance the use of tables and figures?

Response 12: Thank you for comments. Please refer to our response to point 9.

Point 13:

I would suggest not to use citations older than 2015. In some cases, if it is damn necessary then try to use some classical ones. There are multiple references used from 1974, 1982, and 1991, etc. Please ponder them and consider replacement with the latest ones as much as possible.

Response 13: Thank you for the comments. We did retrieve most of the related articles. The reasons for citing these articles were the contributions they had done for the related research.

Reviewer 2 Report

After reading the article I send my appreciation: - the theme is current and timely, contributing to the study of a real problem in the personal and professional life of most people (daily emotional exhaustion and burnout); - the authors adopt a clear and legible technical expression; - the methodology is precise and well defined; - the results are consistent with the research developed; - the conclusions are objective; However, some bibliographic references are very old and their specificity in the study is not evident. It is suggested that they be updated or that their specificity and relevance be highlighted in relation to the study carried out. It is also suggested to revise the text to correct minor errors, such as, for example, line 189 - the residuall vriance - or line 190 - of emmtional.

Author Response

Reviewer 2

Comments and Suggestions for Authors

After reading the article I send my appreciation: - the theme is current and timely, contributing to the study of a real problem in the personal and professional life of most people (daily emotional exhaustion and burnout); - the authors adopt a clear and legible technical expression; - the methodology is precise and well defined; - the results are consistent with the research developed; - the conclusions are objective;

Response: Thank you for comments.

Point 1:

However, some bibliographic references are very old and their specificity in the study is not evident. It is suggested that they be updated or that their specificity and relevance be highlighted in relation to the study carried out.

Response 1:

Thank you for the sincere comments. We did retrieve most of the related articles. The reasons for citing these articles were the contributions they had done for the related research.

Point 2:

It is also suggested to revise the text to correct minor errors, such as, for example, line 189 - the residual vriance - or line 190 - of emmtional.

Response 2:

Thank you for the comment. We had correct errors in the paper. Please refer to line 190 and line 191.

Reviewer 3 Report

Thank you for letting me review this manuscript. This is an interesting paper on a theme of great current interest and about which researchers and society in general require more information.

I have some suggestions that may improve the manuscript.

Please, explain why is necessary to perform this study among employees in health and fitness clubs.

What type of epidemiological study has been carried out?

How were people invited to participate?

Regarding the participants, has any exclusion criteria been considered?

Explain how the study size arrived at the final number. Report numbers of individuals at each stage of study—eg numbers potentially eligible, excluded because of incompleteness, and so on.

Specify the power of the study. Is the sample studied representative of the employees in healthcare and fitness clubs?

Authors should compare and explain their results with papers referred mainly to the study populations. Some references are referred to undergraduates (33. Ambrona et al., 35. Finch et al.), teachers (49. Simbula) or stroke patients (38. Powell et al.).

Author Response

Reviewer 3

Comments and Suggestions for Authors

Thank you for letting me review this manuscript. This is an interesting paper on a theme of great current interest and about which researchers and society in general require more information.

I have some suggestions that may improve the manuscript.

Response: Thank you for your comments.

Point 1:

Please, explain why is necessary to perform this study among employees in health and fitness clubs.

Response 1:

Thank you for comments. In Taiwan, overweight/obese is prevalent. With that, the government encourages people to do exercises to prevent it. Therefore, health and fitness clubs become very popular recently. It concerns us to do such research to understand their needs and may provide practical implications to business runners to better manage their employees.

Point 2:

What type of epidemiological study has been carried out?

Response 2:

Thank you for comments. This study focused on studying employees’ psychological perceptions rather than epidemiological study.

Point 3:

How were people invited to participate?

Response 3:

Sampling method of this study was convenience sampling. Eighty percent of sports clubs are located in six major metropolitan cities in Taiwan. Therefore, we selected 13 clubs’ employees to participate in this study and we believe the sample can represent the population mostly and reduced biasness. (The participants got 30 USD reward for participation)

Point 4:

Regarding the participants, has any exclusion criteria been considered?

Response 4:

Thank you for comments. Sampling method of this study was convenience sampling. We recruited health and fitness clubs’ full- time employees as participants with consents.

Point 5:

Explain how the study size arrived at the final number. Report numbers of individuals at each stage of studied numbers potentially eligible, excluded because of incompleteness, and so on.

Response 5:

Thank you for comments. In the beginning, we were able to recruit 200 participants. However, only179 completed 10-day diary records. Therefore, only 179 samples were used for hypothesis testing.

Point 6:

Specify the power of the study. Is the sample studied representative of the employees in healthcare and fitness clubs?

Response 6:

Thank you for comments. At this moment, only Mplus can analyze RDSEM. So far, there is no technique to calculate the power of the study. Please refer to the following paper.

McNeish, D., & Hamaker, E. L. (2019). A Primer on Two-Level Dynamic Structural Equation Models for Intensive Longitudinal Data in Mplus. Psychological Methods. Advance online publication. http://dx.doi.org/10.1037/met0000250

Point 7:

Authors should compare and explain their results with papers referred mainly to the study populations. Some references are referred to undergraduates (33. Ambrona et al., 35. Finch et al.), teachers (49. Simbula) or stroke patients (38. Powell et al.).

Response 7:

Thank you for the comments. The papers mentioned in our article was used as our theoretical background to deduce our hypotheses. However, the diary method involved multi-level variants, those cannot be used to compared with our results. On the other hand, we did comparisons with other studies in the discussion section. Please refer to the discussion section. Thank you.

Round 2

Reviewer 1 Report

This version is much improved. Yet I would urge authors to improve the method parts as much as possible. Look for minor spell checks if any.